# Fabrication of Micro Ultrasonic Powder Molding Polypropylene Part with Hydrophobic Patterned Surface

**DOI:** 10.3390/ma13153247

**Published:** 2020-07-22

**Authors:** Xiong Liang, Yongjing Liu, Jiang Ma, Feng Gong, Yan Lou, Lianyu Fu, Bin Xu

**Affiliations:** 1College of Mechatronics and Control Engineering, Shenzhen University, Shenzhen 518060, China; xliang@szu.edu.cn (X.L.); 1810293044@email.szu.edu.cn (Y.L.); majiang@szu.edu.cn (J.M.); gongfeng@szu.edu.cn (F.G.); louyan@szu.edu.cn (Y.L.); 2Shenzhen Jinzhou Precision Technology Corp., Shenzhen 518116, China; mlyfu@163.com

**Keywords:** micro ultrasonic powder molding, micro-structured pattern, replicated, core insert, wettability

## Abstract

Constructing regular micro-structures with certain geometric characteristics on the surface of the polymer part can obtain some specific functions. Micro ultrasonic powder molding (micro-UPM) is an efficient processing technique for the fabrication of well-filled micro-structured Polypropylene (PP) parts. The micro-structure array on the surface of the core insert was obtained by low speed wire electrical discharge machining (WEDM-LS). PP polymer surfaces with micro-structured patterns were successfully replicated from the core insert surface after micro-UPM. By studying the detailed topography characterizations of micro-structured PP parts, the effects of processing parameters (ultrasonic energy, welding pressure and holding time) on the micro-structured filling show that when PP polymer was formed under the conditions of 1000 J, 115 kPa and 8 s during micro-UPM, well-filled micro-structured parts can be obtained. Besides, without low surface energy coating modification, the water contact angles (WCAs) of micro-structured PP parts increased from 85.3° to 146.8°, indicating that the wettability of the surface can be changed by replicating the micro-structure on PP parts after micro-UPM.

## 1. Introduction

Processing micro-structures (such as micro-groove arrays, cone arrays and so on) on a material surface can give the material special properties, such as hydrophobicity [1], drug delivery [2], antifriction properties [3], optical properties [4] and self-cleaning [5]. Polymer materials are widely used in the preparation of micro-parts and surface structures because of their low cost, improving manufacturing technology, optical properties and chemical resistance [6].

At present, the molding process of thermoplastic polymers mainly includes hot embossing [7] and injection molding [8]. In the hot embossing process, an external heat source is used to heat the mold and soften the polymer, load the pressure and hold it for a while, to make the polymer fill the cavity, and finally cool and demold [9]. However, the process requires repeated heating and cooling of the mold, resulting in low forming efficiency. Besides, the microplastic part with complex geometry features is difficult to be fabricated by the hot embossing process. In injection molding, the polymer needs to be melted firstly by using a heating device, and then the molten polymer was injected into the mold. After the molten polymer is cooled, the plastic part can be obtained. During injection molding, the filling resistance increases, and a frozen layer is prone to forming on the mold surface. Under the effect of the above factors above, it is difficult for the polymer to be filled in the micro-structure. Moreover, the melt is exposed to high temperatures for a long time, which can lead to degradation of the polymer [10].

To investigate the problems of the traditional process, a method for applying ultrasonic waves to the molding of polymer materials was proposed. Ultrasonic molding of polymers has both physical and chemical effects. The chemical effect is mainly to break the molecular chain and reduce molecular weight. The physical effect is to improve the motion of the molecular chain and reduce the elastic tensile strain. Both types of effects contribute to reducing the viscosity of the polymer significantly and facilitate the filling of the material [11,12]. From the experimental results, it can be found that the morphology of the specimen is closely related to the ultrasonic time, plasticizing pressure, and mold temperature [13,14]. Frequency and amplitude are two main factors that need to be considered when studying the ultrasonic molding process. The viscoelastic heating rate increases with the frequency. Larger amplitude gives better filling and geometry accuracy [15]. By optimizing these parameters, the fluidity of the ultrasonic plasticized polymer melts can be enhanced, the filling length can be increased, and the shear viscosity can be significantly reduced [16,17]. Similar results were also obtained by Lou et al. [18]; their experimental results show that increasing the amplitude can significantly improve the filling capacity of the polymer melt. 

In addition, the responses related to the quality of the sample, such as filling capacity, mechanical properties, crystallinity and repeatability, are also important to be considered. By simulating the melt filling process in the cavity, Gao et al. [19] concluded that the melt filling capacity is enhanced under the action of ultrasonic vibration. Dorf et al. [20] calculated the crystallinity of PEEK samples based on DSC analysis. The results show that Crystallization is similar to the raw material. Ferrer et al. [21] manufactured thin-wall plates with polystyrene microchannels by ultrasonic molding technology, verified the repeatability and reproducibility of the technology. Sánchez-Sánchez et al. [22] employed ultrasonic vibration in the injection molding process. Ultra-high molecular weight polyethylene (UHMWPE) powder was melted and filled into the mold cavity under the effect of ultrasonic vibration, and a small-size tensile test specimen with good quality was successfully prepared. Subsequently, they added graphite to UHMWPE to strengthen the tensile strength of the sample [23]. To obtain a microstructural surface on a three-dimensional part, Zhang et al. [24] used lithography technology to fabricate micro-hole arrays on the Ni plate and adopted it as a mold insert for the polyether ether ketone (PEEK) injection molding experiment.

The above studies provide a good reference on the molding of the plastic parts. The micro ultrasonic powder molding (micro-UPM) is an easier and more economical way to mold microplastic parts. In micro-UPM, ultrasonic vibration was applied to the polymer pellets [25,26]. Under the effect of the ultrasonic vibration, friction between the polymer pellets could take place and, thus, the heat can be generated for fabricating the microplastic parts, which has high forming efficiency [27,28]. During micro-UPM, the heating time of polymer material is short, which avoids the degradation of the polymer. In previous research [29], the shape and size of the micro-structured parts prepared by micro-UPM are unable to be accurately controlled because the material chamber also serves as the mold cavity. This technical shortcoming limits the application of micro-UPM in the industrial field.

Focusing on this problem, a combination mold was made in this paper, which is assembled by an upper plate, lower plate, material chamber, runner insert and micro-structured core insert. The material chamber and the material chamber were connected by the channel insert. Then the mold is applied to micro-UPM. Only by processing different micro-structured core inserts, the micro-structured parts with different shapes and sizes can be obtained. The replication rate and surface roughness Ra are the main indicators of molding quality. High replication rate represents good molding quality and low surface roughness represents good surface quality. Therefore, on the premise of ensuring a high replication rate, the goal of this paper is to reduce the surface roughness of micro-structured parts as much as possible. By applying the core insert with an average surface roughness Ra of 1.45 μm in micro-UPM, the micro-structured part with a replication rate of 96.52% and the surface roughness Ra of 0.85 μm can be obtained. Moreover, the surface wettability of micro-structured parts was also studied.

## 2. Experimental

### 2.1. Materials

The raw material was PP (J170H, Lotte Chemical Co., Chung-Nam, Korea). The average particle size is 3.13 mm, the density is 0.91 g/cm^3^ and the melting point range is 150 ℃–170 ℃.

### 2.2. Mold Design

As shown in Figure 1, the combined mold composed of an upper plate, a lower plate, a material chamber, a channel insert and a micro-structured core insert. The material chamber was separated from the mold cavity and the two were connected by the runner insert. The cross-section of the channel was designed to a semicircle. The length and diameter of the channel are 5 mm and 2.5 mm. 

Using 304# stainless steel as raw materials, the core insert with a micro-groove array structure was prepared by the low speed wire electrical discharge machining (WEDM-LS). The overall size of the core insert was 7 mm × 7 mm × 3 mm. The micro-groove arrays in micro-structured core insert were designed to U-shaped (bottom radius of 70 μm and depth of 140 μm), V-shaped (base angle of 90° and depth of 140 μm) and semi-circular (radius of 70 μm and depth of 70 μm). The micro-groove arrays in micro-structured core insert can also be designed into desired shapes, such as triangles, rectangles and circles.

Applying the combined mold in micro-UPM, micro-structured parts with different shapes and sizes (Figure 2) can be obtained only by replacing different micro-structured core inserts. The micro-groove arrays in micro-structured core insert can also be designed into desired shapes, such as triangles, rectangles and circles. In addition, channels with different cross-sectional shapes (such as semi-circular, triangular and rectangular) or channels with unequal-diameters (such as tapered) can also be assembled into the combined mold.

### 2.3. Ultrasonic Molding Equipment and Processing Flow

The micro-UPM experimental platform was an ultrasonic welder (Branson 2000XCT, EMERSON, St. Louis, MO, USA). The output frequency was 20 kHz, the maximum output power was 2500 W and the diameter of the working end of the sonotrode was 10 mm. During the experiment, the ultrasonic amplitude was set to 80 µm and the ultrasonic welder was operated under the energy control mode.

The preparation process of the micro-structured part is shown in Figure 3. (1) Through WEDM-LS (Figure 3a), the core insert was processed (Figure 3b). (2) The mold was assembled (Figure 3c), and the assembled mold was fixed on the ultrasonic welder (Figure 3d). (3) A certain weight of PP pellets was filled into the material chamber (Figure 3e). The ultrasonic energy (parameter A, unit J), welding pressure (parameter B, unit kPa) and holding time (parameter C, unit s) were set. (4) The ultrasonic welder was started (Figure 3f). Without the external heat source, PP pellets were melted rapidly under the action of ultrasonic vibration. Then, under the action of welding pressure, the melt passed through the channel and filled into the mold cavity (Figure 3g). After that, the pressure was maintained for a certain time to complete the molding of the micro-structured parts (Figure 3h). (5) After cooling the melt, opening the upper plate, the micro-structured part can be taken out from the combined mold (Figure 3i).

### 2.4. Experimental Method

The experiment was divided into two stages. In the first stage, preliminary molding experiments were performed to determine the approximate range of process parameters. In the second stage, a core insert with U-shaped micro-groove array was selected to study the influences of the ultrasonic energy (J), welding pressure (kPa) and holding time (s) on the molding quality of the micro-structured parts. The experimental parameters are shown in Table 1.

### 2.5. Measurement

The surface morphology and cross-sectional area of micro-structured parts were obtained by a laser scanning confocal microscope (LSCM, VK-X250K, Keyence, Osaka, Japan). To reduce error, five micro-structured parts were prepared under each group of process parameters and three different positions were randomly selected in each micro-structured part to measure their sectional areas. The replication rate was calculated by the ratio between the cross-sectional area of the micro-structured part and the cross-sectional area of the micro-structured core insert. The roughness, Ra, was measured by LSCM at the bottom of the micro-groove of the mold core insert and the top of the corresponding micro-structured parts (Figure 4). Each Ra value was measured 10 times in the same place and then the measurement data were averaged to obtain the final Ra value.

The static water contact angle (WCA) of the sample surface was obtained by a drop shape analyzer (DSA100S, Krüss, Hamburg, Germany). Each sample was measured under the same environmental conditions and no low surface energy material modifications were applied to the sample surface. Three different locations for each sample were measured, with the mean value as the final data.

## 3. Results and Discussion

### 3.1. Effect of Ultrasonic Energy on the Molding Quality

Under the effect of the ultrasonic vibration, friction between the polymer pellets could take place, and thus the heat was generated for fabricating the micro-structured part. Therefore, ultrasonic energy has the same effect on mold temperature and different ultrasonic energies represent different mold temperatures. A core insert with a U-shaped micro-groove array was selected as mold. The welding pressure was fixed at 100 kPa and the holding time was fixed at 8 s. The micro-UPM was performed under different ultrasonic energy and the experimental results are shown in Figure 5. When the ultrasonic energy was 400 J, most of the polymers were only locally melted and compressed bonded together in the material chamber. A small amount of polymer melt was filled into the mold cavity through the channel. As a result, the mold cavity was not filled and the U-shaped micro-grooves were not completely replicated. As shown in Figure 5a, the surface defects of the mold parts were obvious and the molding quality was poor. According to the measurement results, the surface roughness Ra of the micro-structured was 3.49 μm and the replication rate was only 53.65% (Table 2).

With the ultrasonic energy increased, the temperature increased to the melting point of the polymer and, thus, more polymers were melted in the material chamber. Under this circumstance, the viscosity of the melt reduced, and the filling capability of the melt was improved. As a result, the overall shape and surface micro-structure were both better replicated. When the ultrasonic energy was 1000 J, the mold cavity was completely filled with the melt and the U-shaped micro-grooves were completely replicated. According to the measurement results, the surface roughness, Ra, of the micro-structured part was 0.87 μm and the replication rate was increased to 94.57% (Table 2). When the ultrasonic energy increased to 1200 J, the replication rate and roughness did not change much. Therefore, an appropriate increase in ultrasonic energy can improve the replication rate and the surface quality of the micro-structured parts. To guarantee the molding quality of micro-structured parts, the ultrasonic energy was set as 1000 J in this paper.

### 3.2. Influence of Welding Pressure on Molding Quality

The ultrasonic energy was selected to be 1000 J, which gave the best molding quality in the previous experiment. The holding time was 8 s. The micro-UPM was performed under different welding pressure and the experimental results are shown in Figure 6.

Under the effect of the welding pressure, the melt obtained a certain flow speed, which made the melt fill into the mold cavity through the channel. Under the above process parameters, the mold cavity was completely filled and the U-shaped groove was replicated. As shown in Figure 6 and Table 3, when the welding pressure was low, the melt did not fill the bottom of the U-shaped groove completely. The replication rate and surface quality of the micro-structured part were poor. With the welding pressure increased, the flowability of the melt was improved, which resulted in better replication of the micro-grooves. When the welding pressure was 115 kPa, the replication rate of the micro-structure was 96.52% and the surface roughness, Ra, was 0.85 μm. However, when the welding pressure was 130 kPa, the excessive welding pressures could cause the filling speed to become too fast. Under this circumstance, the micro-structured part was prone to generate flash and warping, which has an adverse effect on the quality of the micro-structured part. To guarantee the quality of the micro-structured part, the welding pressure was set as 115 kPa in this paper.

### 3.3. Influence of Holding Time on the Molding Quality

After the melt fills the mold cavity, the pre-set pressure needs to be maintained for a certain period of time to compensate for the shrinkage of the parts due to cooling. This time is called the holding time. The ultrasonic energy was 1000 J, and the welding pressure was 115 kPa. The micro-UPM was performed under different holding times and the experimental results were shown in Table 4. As shown in Figure 7, with the holding time increased, the replication rate increased. However, when the holding time was greater than 4 s, the replication rate did not change significantly.

If the holding time was set to 0 s, the pressure of the melt in the material chamber dropped rapidly with the sonotrode reset. Because of the existence of the channel, the small opening of the channel blocked the backflow of the melt in the mold cavity, which was beneficial to fabricate the micro-structured part. After setting a certain holding time, under the pre-set pressure, the melt filled the micro-structured core insert completely, and then it was cooled to form the micro-structured part. With the holding time further increased, the melt flowed into the mold cavity under the pre-set pressure and filled the U-shaped micro-grooves in the initial stage. There is little difference in the replication rate and roughness of the micro-structured parts when the holding time is 4 s and 8 s, respectively. It can be concluded that when the holding time is 4 s, the melt inside the cavity is filled until the holding time is increased to 8 s. The micro-structured parts have been solidified. Thus, the preset pressure directly affects the material in the material chamber and has no effect on the micro-structured parts in the molding chamber after the holding time exceeds 8 s. Based on results above, it can be concluded that the holding time plays a crucial role in the unsolidified stages of the melt. Compared with that of the micro-structure fabricated without any holding time, the roughness of the micro-structure was reduced by 37.96% and the replication rate was increased by 13.93% when the holding time was 8 s. Since the material cavity of the mold is separated from the molding cavity, the holding pressure does not affect the solidified micro-structured parts inside the mold cavity. In order to guarantee the molding efficiency and quality of the micro-structured part, the holding time was set as 8 s in this paper.

In summary, under ultrasonic energy of 1000 J, welding pressure of 115 kPa and holding time of 8 s, micro-structured parts prepared by micro-UPM had a well replication rate of 95.71% and a roughness Ra of 0.87 μm. when core inserts with semi-circular and V-shaped micro-grooves were used, micro-structured parts with good quality could also be fabricated with the above parameters. The average Ra of the surface micro-structure was 0.93 μm and 0.91 μm, respectively. The micro-structure replication rates were 94.45% and 96.18%, respectively. A profile comparison between micro-structured part and the core insert is shown in Figure 8. Applying the combined mold to the micro-UPM, micro-structured parts of different shapes and sizes can be fabricated simply by changing the core insert.

### 3.4. Wettability Properties

The WCAs of the micro-structured parts were measured by a droplet shape analyzer, and the wettability was evaluated according to the measurement results. To investigate the structural anisotropy, droplets were observed in two different observation views, perpendicular view and parallel view (Figure 9). The measurement results of the WCA on the surface of the micro-structured parts are shown in Figure 10. The WCAs in the perpendicular view are all higher than those in the parallel view, and the maximum WCA in the perpendicular view is 146.8°. Besides, with the increase of replication rate of micro-structured parts, the WCAs tended to increase too; this result is particularly evident in the energy group. The WCAs of samples with higher replication rates tends to be stable.

Table 5 shows the WCA optical photos on the structure-free surface and micro-structured parts surface. The WCAs on the structure-free surface and samples 1#–5# in perpendicular view are 85.3°, 119.3°, 127.6°, 125.4°, 141.3° and 140.48°, respectively. The WCAs on the structure-free surface and samples 1#–5# in parallel view are 84.9°, 107.8°,117.4°, 123.7°, 124.7° and 120.2°, respectively. Regardless of the observation views, the WCAs of the structure-free surface are less than 90° in both directions, indicating that polypropylene is a hydrophilic polymer material. Compared to the structure-free surface, the WCAs of the micro-structured parts are all more than 90°, indicating that replicating the microstructure by the micro-UPM process successfully improves hydrophobicity. The anisotropic wettability exhibited on the surface of the micro-structure parts is mainly attributed to the direction and geometric parameters of the micro-structure.

Two classical theories have been proposed to explain the state of droplets on solid surfaces, one of which is the Wenzel model [30] where droplets fill the surface of the micro-structure, and the WCA is expressed by the following formula:(1)$$cos\theta_{W}=rcos\theta_{Y},$$
where r is the roughness factor, and $\theta_{Y}$ is the intrinsic WCA on a structure-free surface. The other state is the Cassie–Baxter model [31] that droplets cannot fill the surface of the micro-structure with air in the gap between the micro-structures, and the WCA is expressed by the following formula:(2)$$cos\theta_{CB}=f\left(1+cos\theta_{Y}\right)-1$$
where f is the fractions of the solid surface in contact with the liquid. Due to the low replication rate and the small filling depth of the micro-structured on sample 1#, the droplets overcome the energy barrier and immerse itself completely into the micro-structure gap as a result of gravity. The droplet was completely glued to the surface, showing the Wenzel state (Figure 11a). Likewise, for samples 2#–5#, with the increase the replication rate, the filling depth of the micro-structured also increased. The droplet was supported by the micro-structures and an interface composed of solid, liquid and gas was formed below it. In this case, the water droplet was less likely to immerse itself into the gap, and the droplet changes from Wenzel state to Cassie–Baxter state (Figure 11b). Figure 12 shows the relationship between the WCAs in the perpendicular view of the sample and the droplet deposition time. The results show that the WCAs in the perpendicular view decreases slowly. The droplets diffusing in the perpendicular direction repeatedly encounter the energy barrier formed by the micro-structure. The energy barrier hindered the diffusion of droplet and the WCAs finally stabilizes [32].

## 4. Conclusions

The micro-structured core insert with regular micro-groove array structure processed by WEDM-LS was applied to micro-UPM, and the micro-structure was replicated using polypropylene. This work studies the effect of micro-UPM on the processability of polypropylene under different process parameters. The primary findings are presented as follows:(1)A combined mold was assembled with an upper plate, a lower plate, a material chamber, a channel insert and a micro-structured core insert. Applying the combined mold in micro-UPM, the micro-structured parts with different shapes and sizes can be prepared, which has a simple process, low production cost and good flexibility.(2)The effect of ultrasonic energy, welding pressure and holding time was investigated for the fabricating of micro-structured PP parts during micro-UPM. The viscosity on the PP polymer was lower when ultrasonic energy is strengthened, leading to it filling in the micro-groove arrays more. Higher welding pressure allows PP melt to fill at a faster rate, reducing heat loss during the filling process. Besides, holding time is one of the critical parameters to prevent melting reflux and reduce product shrinkage and deformation. At the conditions of 1000 J, 115 kPa and 8 s, micro-structured parts with flexible shape and size can be successfully fabricated, the replication rate reaches about 96.52% and roughness Ra is about 0.85 μm.(3)By using the micro-structured core insert processed by WEDM-LS, micro-UPM successfully produced micro-structured parts with hydrophobic surface. The WCA in the perpendicular view on the micro-structured parts can reach a maximum of 146.8°.(4)The conclusions above are drawn from different trials, which proves the reproducibility of the molding process. However, more research on micro-UPM, such as the fabrication of composite materials, is still needed in the future.

## Figures and Tables

**Figure 1 materials-13-03247-f001:**
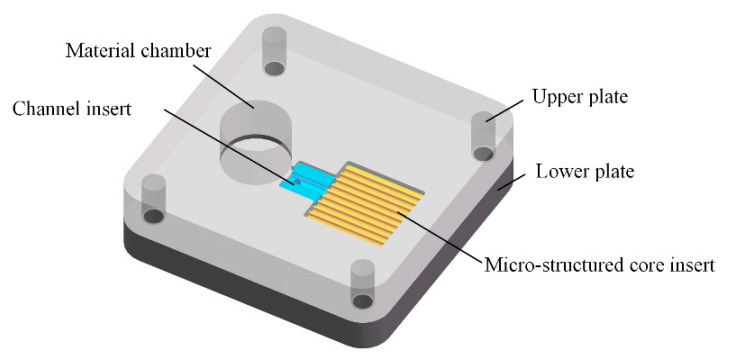
Combined mold.

**Figure 2 materials-13-03247-f002:**
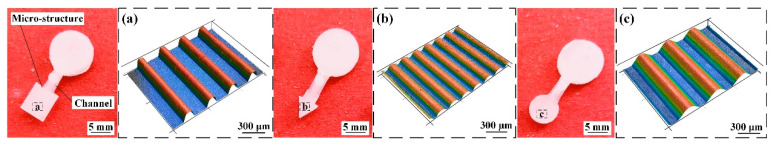
Micro-structured parts and surface morphology: (**a**) U-shaped; (**b**) semi-circular; (**c**) V-shaped.

**Figure 3 materials-13-03247-f003:**
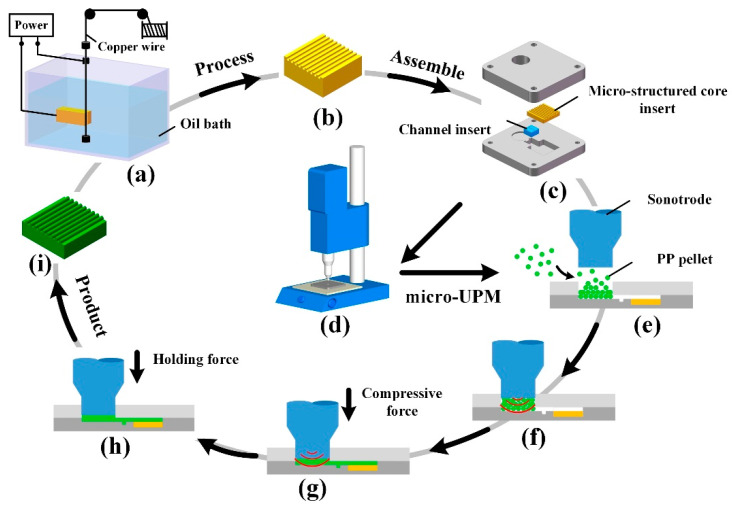
Schematic diagram of the preparation process of the micro-structured parts: (**a**) processing the core insert by the low speed wire electrical discharge machining (WEDM-LS); (**b**) micro-structured core insert; (**c**) assembling the combined mold; (**d**) ultrasonic welder; (**e**) filling with Polypropylene (PP) pellets; (**f**) turning on ultrasonic vibration; (**g**) filling of PP melt into the mold cavity; (**h**) holding state; (**i**) micro-structured part.

**Figure 4 materials-13-03247-f004:**
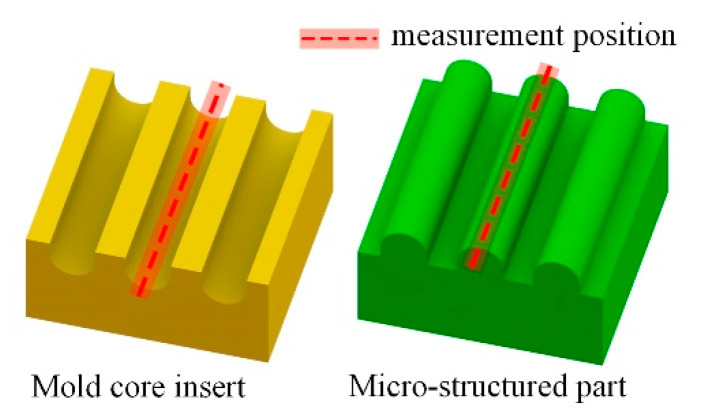
The measurement of roughness (Ra) on the bottom of the micro-groove of the mold core insert and the top of the corresponding micro-structured part.

**Figure 5 materials-13-03247-f005:**
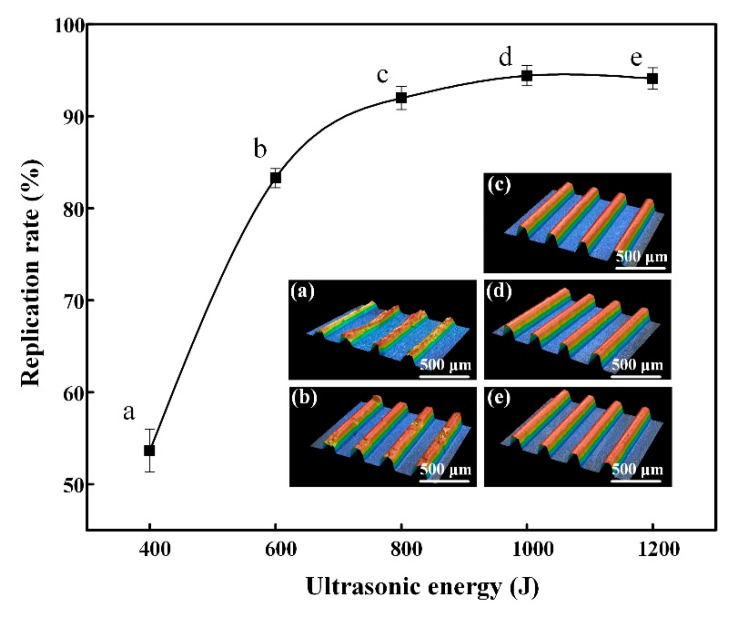
The replication rate of micro-structure at different ultrasonic energies and three-dimensional profile: (**a**) 400 J; (**b**) 600 J; (**c**) 800 J; (**d**) 1000 J; (**e**) 1200 J.

**Figure 6 materials-13-03247-f006:**
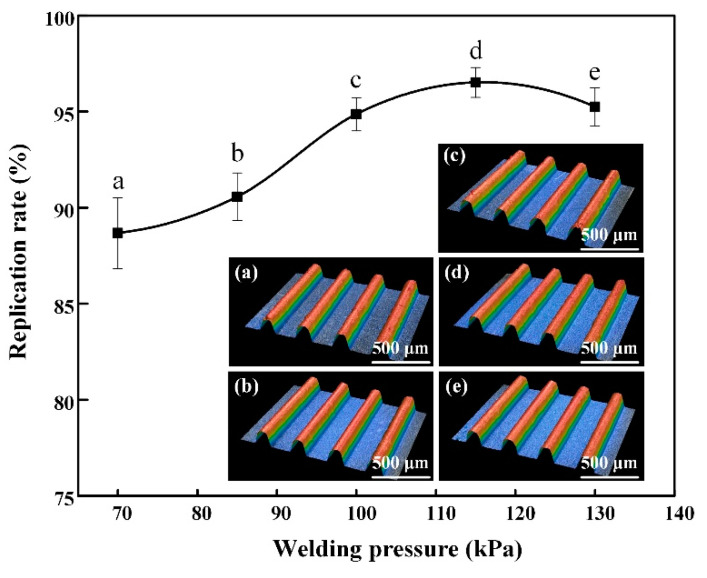
The replication rate of micro-structure at different welding pressures and three-dimensional profile: (**a**) 70 kPa; (**b**)85 kPa; (**c**) 100 kPa; (**d**) 115 kPa; (**e**) 130 kPa.

**Figure 7 materials-13-03247-f007:**
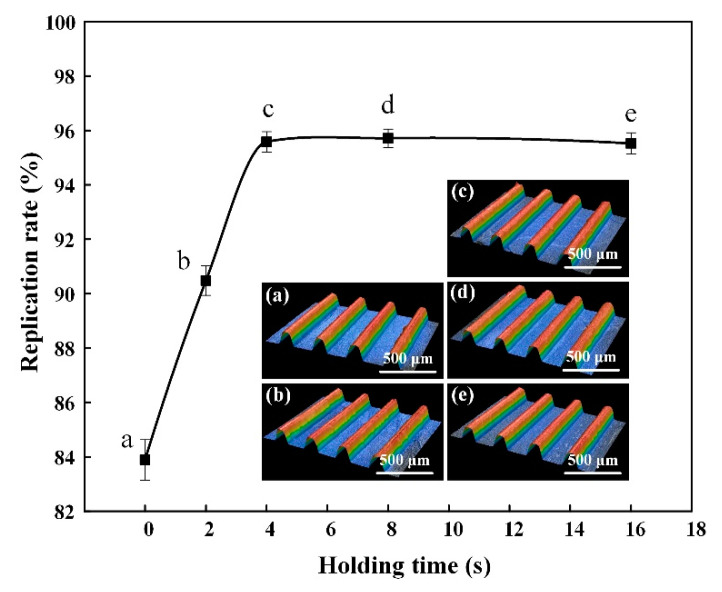
The replication rate of micro-structure at different holding times and three-dimensional profile: (**a**) 0 s; (**b**)2 s; (**c**) 4 s; (**d**) 8 s; (**e**) 16 s.

**Figure 8 materials-13-03247-f008:**
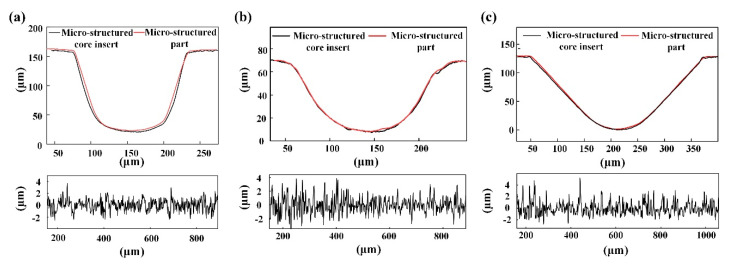
Cross-sectional profiles and roughness curves of the micro-structure: (**a**) U-shaped; (**b**) semi-circular; (**c**) V-shaped.

**Figure 9 materials-13-03247-f009:**
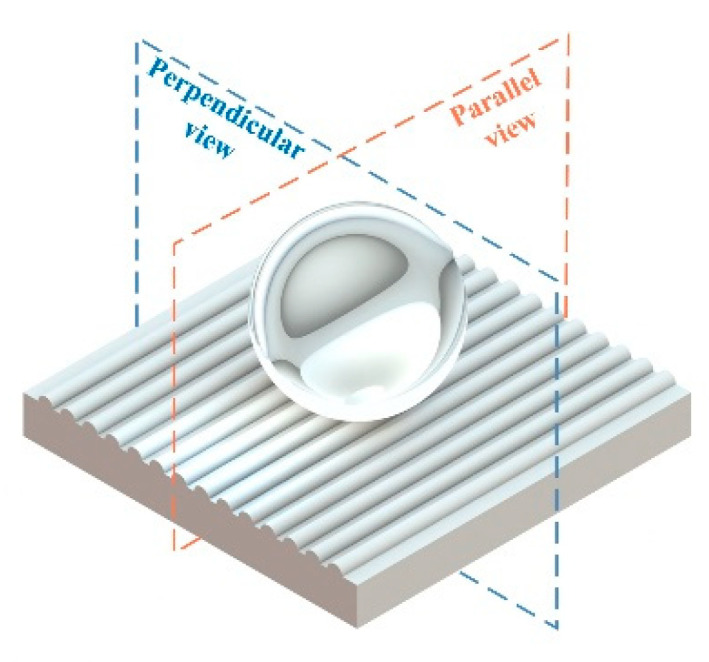
Schematic diagram of measurement direction.

**Figure 10 materials-13-03247-f010:**
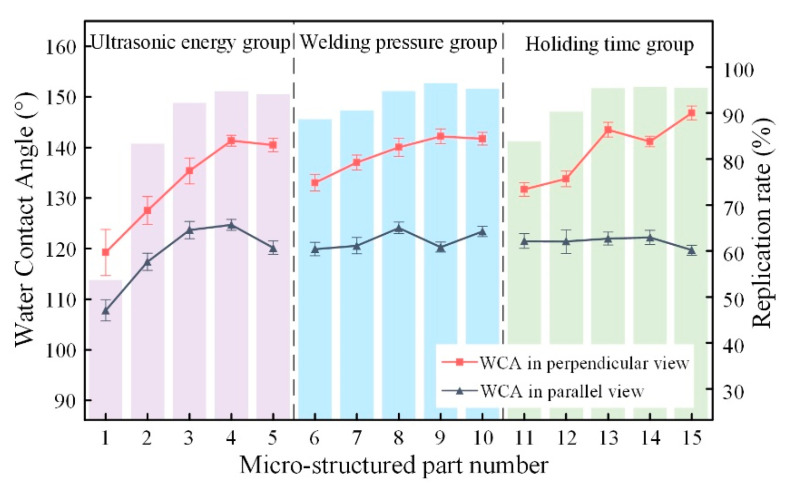
The water contact angles (WCA) and replication rate of micro-structured parts.

**Figure 11 materials-13-03247-f011:**
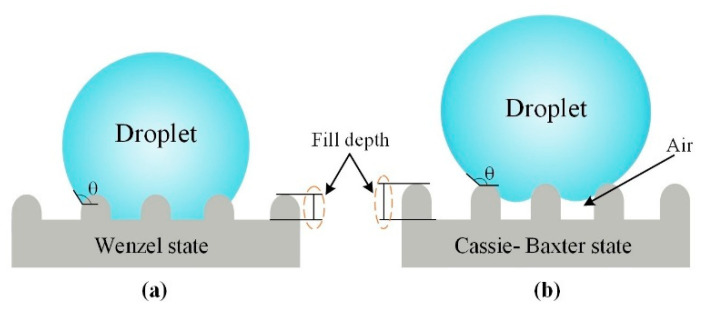
Droplet wettability state at different filling depths of micro-structures: (**a**) Wenzel state; (**b**) Cassie-Baxter state.

**Figure 12 materials-13-03247-f012:**
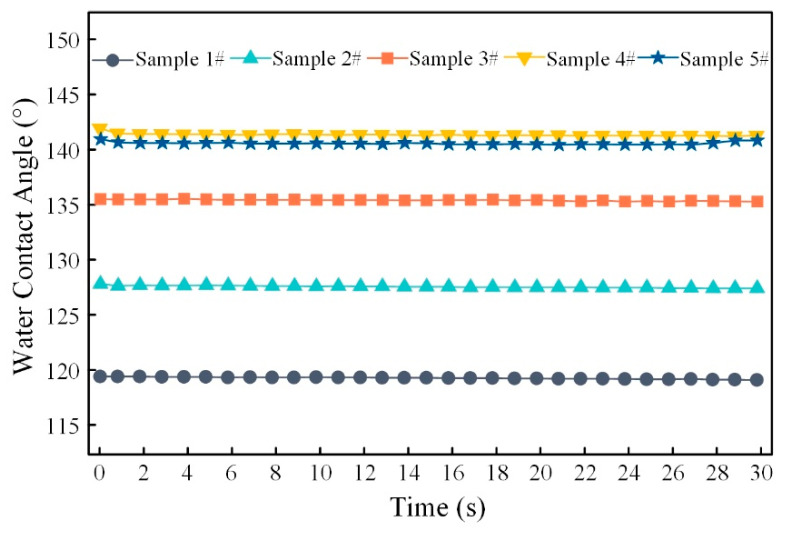
The evolutions of WCAs versus deposition time for PP micro-structured part surfaces.

**Table 1 materials-13-03247-t001:** Process parameters used in experiments.

Process Parameters	Experiment 1	Experiment 2	Experiment 3
A: Ultrasonic energy (J)	400–1200	1000	1000
B: Welding pressure (kPa)	100	70–130	115
C: Holding time (s)	8	8	0–16

**Table 2 materials-13-03247-t002:** Replication rate and surface roughness of micro-structured part fabricated under different ultrasonic energies.

Sample	Ultrasonic Energy (J)	Replication Rate (%)	Surface Roughness (μm)
1#	400	53.65	3.49
2#	600	83.27	1.49
3#	800	92.10	0.91
4#	1000	94.57	0.87
5#	1200	94.08	0.86

**Table 3 materials-13-03247-t003:** Replication rate and surface roughness of micro-structured part fabricated under different welding pressures.

Sample	Welding Pressure (kPa)	Replication Rate (%)	Surface Roughness (μm)
6#	70	88.67	1.02
7#	85	90.56	0.95
8#	100	94.86	0.88
9#	115	96.52	0.85
10#	130	95.23	0.90

**Table 4 materials-13-03247-t004:** Replication rate and surface roughness of micro-structured part fabricated under different holding times.

Sample	Holding Time (s)	Replication Rate (%)	Surface Roughness (μm)
11#	0	83.87	1.37
12#	2	90.47	1.12
13#	4	95.57	0.88
14#	8	95.71	0.85
15#	16	95.55	0.88

**Table 5 materials-13-03247-t005:** WCAs on the surfaces of micro-structured parts.

	Structure-Free Surface	Sample 1#	Sample 2#	Sample 3#	Sample 4#	Sample 5#
Perpendicular view	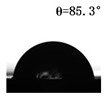	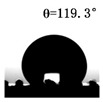	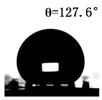	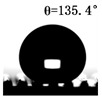	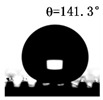	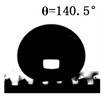
Parallel view	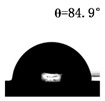	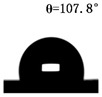	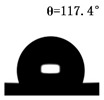	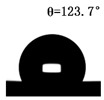	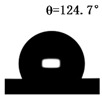	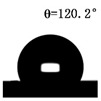

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
