# Peer review of "Fabrication of Micro Ultrasonic Powder Molding Polypropylene Part with Hydrophobic Patterned Surface"

_materials, 2020, doi:10.3390/ma13153247_

Round 1

Reviewer 1 Report

In my opinion the results presented in the manuscript lacks novel results. The results presented here are very obvious to any person aware of the Ultrasonic Powder Molding technology.

The manuscript is well written. It has provided a good introduction to micro-UPM and advantages over different molding techniques. I am impressed by the author method of performing the experiment such as, starting with variation of acoustic energy, followed by variation in welding pressure and later variation in holding time.  I also feel the author method of choosing surface roughness as a parameter to measure quality of the fabricated parts is very good and all agrees with the result. He has provided valid convincing explanation when result starts to deviate.   

But, when I read the citing articles, such as Janer, M.; Planta, X.; Riera, D. Ultrasonic moulding: Current state of the technology. Ultrasonics 2020,102, 14, to me as reader, this manuscript do not tells me new knowledge.  The manuscript lacks the information such providing new knowledge. The author can provide new tests such as effect on density and strength of fabricated products to convey more new information or how fast part can be fabricated. 

Author Response

Reviewer #1: In my opinion the results presented in the manuscript lacks novel results. The results presented here are very obvious to any person aware of the Ultrasonic Powder Molding technology.

The manuscript is well written. It has provided a good introduction to micro-UPM and advantages over different molding techniques. I am impressed by the author method of performing the experiment such as, starting with variation of acoustic energy, followed by variation in welding pressure and later variation in holding time. I also feel the author method of choosing surface roughness as a parameter to measure quality of the fabricated parts is very good and all agrees with the result. He has provided valid convincing explanation when result starts to deviate.

But, when I read the citing articles, such as Janer, M.; Planta, X.; Riera, D. Ultrasonic moulding: Current state of the technology. Ultrasonics 2020,102, 14, to me as reader, this manuscript do not tells me new knowledge. The manuscript lacks the information such providing new knowledge. The author can provide new tests such as effect on density and strength of fabricated products to convey more new information or how fast part can be fabricated.

Author response: Thanks for your suggestion. The suggestion you mentioned about adding some new tests, such as testing the density and strength of micro-structured parts, is a really good idea to provide some new information. More testing work on micro-structured parts will be considered in our future research. In this study, we investigated the effect of ultrasonic energy, welding pressure, and holding time on the molding of micro-structured parts. Replication rate and roughness are mainly used as quality parameters of the micro-structured parts. Finally, the wettability of the micro-structured parts fabricated under different parameters was obtained. Based on your suggestions, we have made changes in the manuscript to better present some of the information we have obtained. In the abstract, we revised: “Constructing regular micro-structures with certain geometric characteristics on the surface of the polymer part can obtain some specific functions. Micro ultrasonic powder molding (micro-UPM) is an efficient processing technique for the fabrication of well-filled micro-structured Polypropylene (PP) parts. The micro-structure array on the surface of the core insert was obtained by low speed wire electrical discharge machining (WEDM-LS). PP polymer surfaces with micro-structured patterns were successfully replicated from the core insert surface after micro-UPM. By studying the detailed topography characterizations of micro-structured PP parts, the effects of processing parameters (ultrasonic energy, welding pressure and holding time) on the micro-structured filling show that when PP polymer was formed under the conditions of 1000 J, 115 kPa, and 8 s during micro-UPM, well-filled micro-structured parts can be obtained. Besides, without low surface energy coating modification, the water contact angles (WCAs) of micro-structured PP parts increased from 85.3° to 146.8°, indicating that the wettability of the surface can be changed by replicating the micro-structure on PP parts after micro-UPM.” Thanks for your useful comment again.

Reviewer 2 Report

The paper analysis the fabrication of micro ultrasonic powder molding  polypropylene part with hydrophobic patterned surface.

From the analysis of the information presented in the article, I found the following:

- the paper presents a series of results that could be of interest to the scientific community;

- the research methodology is unclear and rudimentary;

- the whole research is based on the adjustment of three parameters used to molding polypropylene part;

according to those presented, in the experimental research, the values for two parameters were kept constant and the value for one parameter was varied. This research methodology is not acceptable because it is not possible to optimize the values for the three parameters in order to establish the working regime that would allow obtaining the best parts. A more complex research program (factorial experiences, etc.) should be used that takes into account different values for the three parameters;

- Regarding the ultrasounds, only one parameter was adjusted, namely the ultrasonic energy, but another very important parameter should be taken into account, namely the frequency of the ultrasounds. The paper only specifies that the device used can generate ultrasound with a frequency of 20 kHz<

- the discussion part is rudimentary and does not compare the results obtained with results presented in other scientific papers;

- the conclusions are quantitative and without presenting a possibility to use the results obtained in practice;

- the future research directions are not presented.

Thus, the article cannot be accepted for publication in this form.

Author Response

Reviewer #2, Concern # 1: The paper analysis the fabrication of micro ultrasonic powder molding polypropylene part with hydrophobic patterned surface.

From the analysis of the information presented in the article, I found the following:

The paper presents a series of results that could be of interest to the scientific community;

The research methodology is unclear and rudimentary;

The whole research is based on the adjustment of three parameters used to molding polypropylene part;

According to those presented, in the experimental research, the values for two parameters were kept constant and the value for one parameter was varied. This research methodology is not acceptable because it is not possible to optimize the values for the three parameters in order to establish the working regime that would allow obtaining the best parts. A more complex research program (factorial experiences, etc.) should be used that takes into account different values for the three parameters;

Author response: Thanks for your comment. Systematic methodology is indeed a more efficient method to analyze the impact of multiple factors. In the case of a small number of parameters, single-factor experiments can better present the specific effects of a parameter. Through mathematical calculation and the adjustments of the parameters, the molding process can be well observed. After reading the cited references (Jiang et al. Polymers 2019, 11, 357 and Sacristan et al. Ultrason Sonochem. 2014, 21, 376-386) we decided to design a single-factor experiment to study the effect of process parameters on the molding of micro-structured parts. A more complex research program will be considered to study the interaction of multiple parameters in our future research. Thanks for your useful comment again.

Reviewer #2, Concern # 2: Regarding the ultrasounds, only one parameter was adjusted, namely the ultrasonic energy, but another very important parameter should be taken into account, namely the frequency of the ultrasounds. The paper only specifies that the device used can generate ultrasound with a frequency of 20 kHz.

Author response: Thanks for your suggestion and sorry for our negligence. As you mentioned, frequency is indeed an important parameter of ultrasound. The maximum viscoelastic heating rate increases with the increase of ultrasonic frequency. However, in the ultrasonic welder (2000XCT, Branson, USA), all components must be operated at the same frequency. In this case, the ultrasonic generator and transducer are only capable of delivering one resonant frequency 20 kHz.

Thanks for your useful comment again.

Reviewer #2, Concern # 3: The discussion part is rudimentary and does not compare the results obtained with results presented in other scientific papers;

Author response: Thanks for your comment. Although many researchers have made various efforts, due to the use of different molding devices, the influence of process parameters such as ultrasonic energy, welding pressure, and holding time on the replication rate, roughness, and other responses are not exactly the same. For the comparison of molding methods, we updated the manuscript by modifying the corresponding part. In section 3.3, we revised: “Since the material cavity of the mold is separated from the molding cavity, the holding pressure does not affect the solidified micro-structured parts inside the mold cavity…Applying the combined mold to the micro-UPM, micro-structured parts of different shapes and sizes can be fabricated simply by changing the core insert.” Thanks for your useful comment again.

Reviewer #2, Concern # 4: The conclusions are quantitative and without presenting a possibility to use the results obtained in practice; the future research directions are not presented.

Author response: Thanks for your suggestion and sorry for our negligence. Based on your useful comment, we updated the manuscript by modifying the corresponding part. In conclusion, we revised: “(2) The effect of ultrasonic energy, welding pressure, and holding time was investigated for the fabricating of micro-structured PP parts during micro-UPM. The viscosity on the PP polymer was lower when with ultrasonic energy is strengthened, leading to more filling in the micro-groove arrays. Higher welding pressure allows PP melt to fill at a faster rate, reducing heat loss during the filling process. Besides, the holding time is one of the critical parameters to prevent melting reflux and reduce product shrinkage and deformation. At the conditions of 1000 J, 115 kPa and 8 s, micro-structured parts with flexible shape and size can be successfully fabricated, the replication rate reaches about 96.52% and roughness Ra is about 0.85 μm…(4) The conclusion above are drawn from different trials, which proves the reproducibility of the molding process. However, more research on micro-UPM, such as the fabrication of composite materials, is still needed in the future.” Thanks for your useful comment again.

Reviewer 3 Report

This study shows that the influence of ultrasonic energy, welding pressure, holding time on the ultrasonic plasticization of polypropylene in micro ultrasonic powder molding. Overall, this manuscript is well-organized. The results are also useful for micro ultrasonic powder molding. However, the additional data and revision of the title, abstract, and conclusion are necessary to be published. My comments are as follows:

[Title, Abstract, Conclusion] The experiments were conducted with only single type of polypropylene (J170H, Lotte Chemical Co., Korea). The optimal conditions obtained in this experiment are considered valid only for the specific material. However, the title and conclusion seemed to obtain the optimal process parameters for general polypropylene. Therefore, it would be reasonable to revise the title and the conclusion by emphasizing the results are effective for a specific material with certain ultrasonic conditions.

[Section 2.3.] The frequency and amplitude are critical conditions in ultrasonic vibration. In this study, the vibration frequency and amplitude were 20 kHz and 40 μm, respectively. Were they decided due to the specification limitation of the ultrasonic welder used in this study? Or, it would be better to describe why those conditions were set. Additionally, it would be good to briefly describe the influence of the frequency and amplitude for readers.

[Section 2.5.] Please, describe what kind of equation or method was used to calculate the replication rate.

[Section 3.3.] Compared to welding pressure, the holding time did not increased linearly. If the holding time of two seconds can affect significant surface roughness such as the comparison between the sample 12# and 13#, the interval time of the holding should not be too large such as 4 and 8. The additional experiments for the holding time of 6, 10, 12, and 14 seconds are recommended. At least, the experiment for the holding time of 12 seconds is necessary to support that the holding time of 8 seconds is optimal.

[Line 67] A space between “research” and “[20]” is needed.

Author Response

Reviewer #3, Concern # 1: [Title, Abstract, Conclusion] The experiments were conducted with only single type of polypropylene (J170H, Lotte Chemical Co., Korea). The optimal conditions obtained in this experiment are considered valid only for the specific material. However, the title and conclusion seemed to obtain the optimal process parameters for general polypropylene. Therefore, it would be reasonable to revise the title and the conclusion by emphasizing the results are effective for a specific material with certain ultrasonic conditions.

Author response: Thanks for your comment and we apologize for the confusion to you. In this work, taking polypropylene as an example to study the influence of ultrasonic energy, welding pressure, and holding time on the forming of micro-structured parts. At the conditions of 1000 J, 115 kPa and 8 s, a micro-structured part with a replication rate of 96.52% and a roughness of 0.85 μm is obtained, which provides a reference for micro-UPM of other polymer materials. Based on your useful comment, we updated the manuscript by modifying the corresponding part. In abstract, we revised: “Constructing regular micro-structures with certain geometric characteristics on the surface of the polymer part can obtain some specific functions. Micro ultrasonic powder molding (micro-UPM) is an efficient processing technique for the fabrication of well-filled micro-structured Polypropylene (PP) parts. The micro-structure array on the surface of the core insert was obtained by low speed wire electrical discharge machining (WEDM-LS). PP polymer surfaces with micro-structured patterns were successfully replicated from the core insert surface after micro-UPM. By studying the detailed topography characterizations of micro-structured PP parts, the effects of processing parameters (ultrasonic energy, welding pressure, and holding time) on the micro-structured filling show that when PP polymer was formed under the conditions of 1000 J, 115 kPa and 8 s during micro-UPM, well-filled micro-structured parts can be obtained. Besides, without low surface energy coating modification, the water contact angles (WCAs) of micro-structured PP parts increased from 85.3° to 146.8°, indicating that the wettability of the surface can be changed by replicating the micro-structure on PP parts after micro-UPM.” In conclusion, we revised: “(2) The effect of ultrasonic energy, welding pressure, and holding time was investigated for the fabricating of micro-structured PP parts during micro-UPM. The viscosity on the PP polymer was lower when ultrasonic energy is strengthened, leading to more filling in the micro-groove arrays. Higher welding pressure allows PP melt to fill at a faster rate, reducing heat loss during the filling process. Besides, holding time is one of the critical parameters to prevent melting reflux and reduce product shrinkage and deformation. At the conditions of 1000 J, 115 kPa and 8 s, micro-structured parts with flexible shape and size can be successfully fabricated, the replication rate reaches about 96.52% and roughness Ra is about 0.85 μm.” Thanks for your useful comment again.

Reviewer #3, Concern # 2: [Section 2.3.] The frequency and amplitude are critical conditions in ultrasonic vibration. In this study, the vibration frequency and amplitude were 20 kHz and 40 μm, respectively. Were they decided due to the specification limitation of the ultrasonic welder used in this study? Or, it would be better to describe why those conditions were set. Additionally, it would be good to briefly describe the influence of the frequency and amplitude for readers.

Author response: Thanks for your suggestion and sorry for our negligence. As you mentioned, frequency and amplitude are indeed critical parameters of ultrasound. However, in the ultrasonic welder (2000XCT, Branson, USA), all components must be operated at the same frequency. In this case, the ultrasonic generator and transducer are only capable of delivering one resonant frequency 20 kHz. Processing a material with too lower amplitudes can introduce non-melted material in the cavity  or produce incomplete parts. In general, larger amplitude gives better filling and geometry accuracy, we prefer to fix the amplitude to a higher value to study the influence of other factors. Based on your useful comment, we updated the manuscript by modifying the corresponding part. In the abstract, we revised: “Frequency and amplitude are two main factors that need to be considered when studying the ultrasonic molding process. The viscoelastic heating rate increases with the frequency; Larger amplitude gives better filling and geometry accuracy [14].” Thanks for your useful comment again.

Reviewer #3, Concern # 3: Please, describe what kind of equation or method was used to calculate the replication rate.

Author response: Thanks for your comment and sorry for the confusion. The replication rate is defined as the ratio between the cross-sectional area of the micro-structured part and the cross-sectional area of the core insert. We define the cross-sectional area of the micro-structured part as S1 and the cross-sectional area of the mold core as S2 (figure 1). Therefore, dividing S1 by S2 is the replication rate. All cross-section areas are obtained by laser scanning confocal microscope.

Figure 1 The cross-sectional area of micro-structured part and core insert.

Based on your useful comment, we updated the manuscript by modifying the corresponding part. In section 2.5, we revised: “The replication rate was calculated by the ratio between the cross-sectional area of the micro-structured part and the cross-sectional area of the micro-structured core insert.” Thanks for your useful comment again.

Reviewer #3, Concern # 4: [Section 3.3.] Compared to welding pressure, the holding time did not increase linearly. If the holding time of two seconds can affect significant surface roughness such as the comparison between the sample 12# and 13#, the interval time of the holding should not be too large such as 4 and 8. The additional experiments for the holding time of 6, 10, 12, and 14 seconds are recommended. At least, the experiment for the holding time of 12 seconds is necessary to support that the holding time of 8 seconds is optimal.

Author response: Thanks for your comment and sorry for the confusion. Since ultrasonic energy and welding pressure have been optimized in the previous section, the effect of holding time on polymer molding is limited. As shown in Figure 7 in the manuscript. When the holding time is 4 s and 8 s respectively, the replication rate and roughness of the micro-structured parts are not more different. It can be concluded that when the holding time is 4 s, the melt in the cavity is filled until the holding time increases to 8 s, the micro-structured parts have been solidified. Since the material cavity of the mold is separated from the molding cavity, the holding time does not affect the micro-structured parts inside the molding cavity after it is solidified. Based on your useful comment, we updated the manuscript by modifying the corresponding part. In section 3.3, we revised: “There is little difference in the replication rate and roughness of the micro-structured parts when the holding time is 4 s and 8 s, respectively. It can be concluded that when the holding time is 4 s, the melt inside the cavity is filled until the holding time is increased to 8 s. The micro-structured parts have been solidified…Based on the analysis of the above results, it can be concluded that the holding time plays a crucial role in the unsolidified stages of the melt…Since the material cavity of the mold is separated from the molding cavity, the holding pressure does not affect the solidified micro-structured parts inside the mold cavity.” Thanks for your useful comment again.

Reviewer #3, Concern # 5: [Line 67] A space between “research” and “[20]” is needed.

Author response: Based on your comment, we updated the manuscript by adding a space between “research” and “[20]”. In addition, we checked the manuscript in detail and corrected other errors. Thanks for your useful comment again.

Round 2

Reviewer 2 Report

The authors responded to the made comments in review. The article can be published in the presented form.

Reviewer 3 Report

The authors addressed all my comments carefully.